# Minimizing Travel Time and Latency in Multi-Capacity Ride-Sharing Problems

**Kelin Luo ***[ID] **and Frits C. R. Spieksma** [ID]

Department of Mathematics and Computer Science, Eindhoven University of Technology,
5600 MB Eindhoven, The Netherlands; f.c.r.spieksma@tue.nl
* Correspondence: k.luo@tue.nl

**Abstract:** Motivated by applications in ride-sharing and truck-delivery, we study the problem of matching a number of requests and assigning them to cars. A number of cars are given, each of which consists of a location and a speed, and a number of requests are given, each of which consists of a pick-up location and a drop-off location. Serving a request means that a car must first visit the pick-up location of the request and then visit the drop-off location. Each car can only serve at most $c$ requests. Each assignment can yield multiple different serving routes and corresponding serving times, and our goal was to serve the maximum number of requests with the minimum travel time (called $CS_{sum}$) and to serve the maximum number of requests with the minimum total latency (called $CS_{lat}$). In addition, we studied the special case where the pick-up and drop-off locations of a request coincide. Both problems $CS_{sum}$ and $CS_{lat}$ are APX-hard when $c \geq 2$. We propose an algorithm, called the transportation algorithm (TA), which is a $(2c - 1)$-approximation (resp. $c$-approximation) algorithm for $CS_{sum}$ (resp. $CS_{lat}$); these bounds are shown to be tight. We also considered the special case where each car serves exactly two requests, i.e., $c = 2$. In addition to the TA, we investigated another algorithm, called the match-and-assign algorithm (MA). Moreover, we call the algorithm that outputs the best of the two solutions found by the TA and MA the CA. We show that the CA is a two-approximation (resp. 5/3) for $CS_{sum}$ (resp. $CS_{lat}$), and these ratios are better than the ratios of the individual algorithms, the TA and MA.

**Keywords:** ride-sharing; approximation algorithms; transportation problem





## 1. Introduction

In the multi-capacity ride-sharing problem, we are given a set of cars (or trucks) $D$, each car $k \in D$ located at location $d_k$, and a set of requests $R$, each request $r \in R$ consisting of a source $s_r$ (pick-up location) and a destination $t_r$ (drop-off location). Travel times are given between each pair of locations. Each car $k \in D$ has capacity $c$. Serving a request means that a car first visits the pick-up location of the request (customer or parcel) and then the drop-off location. Each car can serve multiple requests at the same time. This offers the opportunity to share rides, which may reduce the travel time or traffic congestion. This paper is concerned with two objectives when assigning the maximum number of requests ($\min\{|R|, c \cdot |D|\}$ requests): one is to assign requests to the cars such that each car serves at most $c$ requests while minimizing the total travel time, and the other problem is to assign requests to the cars such that each car serves at most $c$ requests while minimizing the total waiting time (called total latency) incurred by customers that have submitted the requests. We now provide more insight into these two objectives:

- **Minimize total travel time:** In this problem, we considered assigning the maximum number of requests to cars, each with no more than $c$ requests, to minimize the total travel time, which is the sum of the travel times a car drives to serve its requests. Viewed from the ride-sharing company or drivers, minimizing the total travel time is the most important, since it results in minimizing the costs while serving the maximum

number of requests. Furthermore, it also results in the minimum pollution or emissions. A solution for a given instance is a collection of trips with the minimum total travel time where a car visits all locations of the requests assigned to that car, while visiting the pick-up location of a request before the corresponding drop-off location. We call the ride-sharing problem with the objective of minimizing the total travel time $CS_{sum}$ and the special case of $CS_{sum}$ where the pick-up and drop-off locations are identical for each request $CS_{sum,s=t}$;

- **Minimize total latency:** In this problem, we considered assigning the maximum number of requests to cars, each with no more than $c$ requests, to minimize the total waiting time, which is the sum of the travel times needed for each individual request (customer or parcel) to arrive at the destination. Passengers or clients care about reaching their destinations as soon as possible. Here, the goal is to obtain a solution that is a collection of trips where the travel time summed over the individual requests is minimum. We call the ride-sharing problem with the objective of minimizing the total latency $CS_{lat}$ and the special case of $CS_{lat}$ where the pick-up and drop-off locations are identical for each request $CS_{lat,s=t}$.

### 1.1. Motivation

Many ride-sharing companies (see [1]) provide a service (carpooling, ride-sharing, etc.) where customers submit their requests and then wait for the company to assign them a car. Consider a large number of requests in a working day morning, each consisting of a pick-up and a drop-off location. The company has a number of available cars whose locations and capacity $c$ are known. The value of capacity $c$ can be seen as the capacity of each car over time, i.e., the number of requests a car can accommodate in a relevant period of time. This value may well differ from the instantaneous capacity of a car (say the number of seats), as a pair of requests served by the same car may not be served simultaneously as one request is dropped off before the other request is picked up. In order to achieve a balanced allocation of requests to cars, each car receives no more than $c$ requests. The task then is to assign the maximum number requests to available cars with respect to the capacity constraint.

It is a fact, however, that in many practical situations, "each request is allowed to occupy at most two seats in a car" (see Uber [1]). A regular vehicle has 4–8 seats; thus, only a limited number of requests can be combined in a single vehicle; this can be modeled by taking $c \leq 4$.

Consider the application of our problem in the area of collective transport. For instance, the company TransVision [2] provides transport service for specific groups of people (patients, commuters, etc.), and they organize collective transport by collecting requests in a particular region of The Netherlands in advance, combine these requests, and assign them to some regular transport companies. To access their service, customers must make their request the evening before the day of the actual transport; the number of requests for a day often exceeds 5000. In this application, each server (car, bus, etc.) may pick up more than four requests during its working period; hence, a value of $c > 4$ can be appropriate.

We can capture the above scenarios by the following problem: There is a set of customers who have specified their pick-up locations and drop-off locations to the vehicle provider, and the provider has a set of cars (also with drivers) that have a specified location and capacity $c$. The task, in this paper, is to assign customers to vehicles without exceeding the vehicles' capacities and plan a service route for each of the vehicles based on optimization criteria, either minimizing the total travel time in $CS_{sum}$ or minimizing the total latency in $CS_{lat}$.

The problems $CS_{sum,s=t}$ and $CS_{lat,s=t}$ are natural special cases of $CS_{sum}$ and $CS_{lat}$, respectively, and can be used to model situations where parcels have to be delivered to clients (whose location is known and fixed). For instance, one can imagine a retailer sending out trucks to satisfy clients' demands where each truck is used to satisfy multiple clients.

*1.2. Related Work*

There is a growing amount of literature related to ride-sharing (see [3] for a survey). In a ride-sharing system, a number of cars are provided to serve requests from customers in a fixed period of time. Typically, there are four types of ride-sharing models: one-to-one, meaning that each car serves a single request at a time (see [4–7]); one-to-many, meaning that each car can serve multiple requests at the same time (see [8,9]); many-to-one, meaning that one request can be served consecutively by multiple cars ([10]); many-to-many, which is a combination of the previous two models ([11]). The ride-sharing problem is to match requests and cars while either minimizing the cost (see [8,9]) or maximizing the profit (see [5–7,10]). In this paper, we study a ride-sharing problem of the one-to-many type with the objective of minimizing the cost.

Different versions of ride-sharing problems have been studied. Alonso-Mora et al. [12] and Pavone et al. [13] estimated what fleet size is appropriate for a city considering the cost of cars, a maximum waiting time for a customer, and the extra expense of moving cars. Agatz et al. [14] studied the problem of assigning cars to customers in real-time to minimize the expected total delivery cost. For a dynamic ride-sharing problem, Stiglic et al. [15] analyzed and showed that a small increase in the flexibility of either the cars or the customers can significantly increase performance. Furthermore, Wang et al. [16] introduced the notion of the stability of a ride-sharing system, and they presented methods to establish stable or nearly stable solutions. Considering the online ride-sharing model, Ashlagi et al. [17] studied the problem of matching requests while they arrive one by one and each of them must be either matched to another request within a prespecified period of time or discarded. Each request can be matched at most once and yields a positive profit. To maximize the total profit while requests arrive in an adversarial model, they provided a randomized four-competitive algorithm. Lowalekar et al. [18] studied a special case of the online version of the ride-sharing problem, such that the vehicles have to return to their depot after serving a number of requests. Guo and Luo [19] designed algorithms for the online ride-sharing problem under both the adversarial model and the random arrival model.

Mori and Samaranayake [20] studied the ride-sharing problem with arbitrary capacity while relaxing the assumption of serving all requests. They used an LP-based randomized rounding algorithm to obtain a solution, such that the expected fraction of unassigned requests was at most $1/e$, while the total cost of serving assigned requests was no more than the optimal solution.

This paper deals with a setting where the maximum number of requests needs to be assigned to the cars such that each car serves no more than $c$ requests while minimizing the total travel time ($CS_{sum}$) or minimizing the total waiting time ($CS_{lat}$). As far as we are aware, this particular ride-sharing problem has not been extensively studied, especially for the latency criterion, i.e., $CS_{lat}$. Notice that when $c = 1$, the ride-sharing problems $CS_{sum}$ and $CS_{lat}$ become minimum weight assignment problems, and an optimal solution can be found in $O(|D|^3)$ (see, e.g., [21]). Bei and Zhang [9] considered $CS_{sum}$ with $c = 2$ and gave a 2.5-approximation algorithm for it. Luo and Spieksma [22] proposed approximation algorithms for four versions of the problem, while still assuming $c = 2$. Here, we generalize the ride-sharing problem to a problem involving any arbitrary constant $c \geq 2$.

In fact, both $CS_{sum}$ and $CS_{lat}$ with $c = 2$ are a special case of the so-called two-to-one assignment problem (2-1-AP) investigated by Goossens et al. [23]. Given a set $G$ of $n$ green elements and a set $R$ of $2n$ red elements, we call a *triple* a set of three elements that consists of a single green element and two red elements. Each triple has a non-negative cost-efficient, and the goal of the 2-1-AP problem is to find a collection of triples such that each element is covered exactly once while minimizing the sum of the corresponding cost coefficients. In the context of our ride-sharing problem with $c = 2$, the green elements represent the cars, and the red elements represent the requests. The arguments presented in [23] allowed us to conclude that both $CS_{sum}$ and $CS_{lat}$ are APX-hard, already for $c = 2$.

For the special case of 2-1-AP where the cost of each triple $(i, j, k)$ is defined as the sum of the three corresponding distances, i.e., $\text{cost}(i, j, k) = d_{ij} + d_{jk} + d_{ki}$, where the distances $d$ satisfy the triangle inequality, Goossens et al. [23] gave an algorithm with the approximation ratio of $4/3$. The definition of the cost coefficients in $\text{CS}_{sum}$, as well as in $\text{CS}_{lat}$ differs from the above expression for $cost(i, j, k)$; we refer to Section 4 for a precise definition.

### 1.3. Our Results

We formulated and analyzed an algorithm, called the transportation algorithm (TA), that outputs a feasible solution to each of the four problems described above. This transportation algorithm belongs to a type of heuristics, called *hub* heuristics, which have been analyzed in the context of the multi-index assignment and multi-index transportation problems (see [24,25]). We identified the worst-case ratios of the TA for the four problems and show them to be tight (see [26] for the appropriate terminology). An overview of these results is shown in Table 1, where "*" means that the corresponding worst-case ratio is tight.

**Table 1.** Overview of our results for ride-sharing problems with $c \geq 2$.

| Problem | TA |
|:---:|:---:|
| $\text{CS}_{sum}$ | $(2c - 1)$ * (Theorem 1) |
| $\text{CS}_{sum,s=t}$ | $(2c - 1)$ * (Theorem 1) |
| $\text{CS}_{lat}$ | $c$ * (Theorem 2) |
| $\text{CS}_{lat,s=t}$ | $c$ * (Theorem 2) |

For the case $c = 2$, we propose a so-called match-and-assign algorithm, the MA. We also define an algorithm, the CA, that consists of outputting the better of the solutions found by the TA and MA. An overview of the results for $c = 2$ is shown in Table 2 (see also [22]). Notice that for $\text{CS}_{sum,s=t}$, $\text{CS}_{lat}$, and $\text{CS}_{lat,s=t}$, the worst-case ratio of the combined algorithm (CA) is strictly better than each of the two worst-case ratios of the individual algorithms f which CA is composed.

**Table 2.** Overview of our results for ride-sharing problems with $c = 2$.

| Problem | MA | TA | CA |
|:---:|:---:|:---:|:---:|
| $\text{CS}_{sum}$ | 2 * (Theorem 3) | 3 * (Theorem 1) | 2 * (Theorem 6) |
| $\text{CS}_{sum,s=t}$ | 1.5 * (Theorem 4) | 3 * (Theorem 1) | 7/5 * (Theorem 7) |
| $\text{CS}_{lat}$ | 2 * (Theorem 5) | 2 * (Theorem 2) | 5/3 (Theorem 8) |
| $\text{CS}_{lat,s=t}$ | 2 * (Theorem 5) | 2 * (Theorem 2) | 3/2 * (Theorem 9) |

The paper is organized as follows. In Section 2, we give a precise problem description. In Section 3, we present the transportation algorithm (TA) and analyze its performance for both $\text{CS}_{sum}$ and $\text{CS}_{lat}$. In Section 4, we consider the special case where each car serves exactly two requests. We propose the match-and-assign algorithm (MA) and analyze the performance of the MA and CA (the better solution of the MA and TA) for both $\text{CS}_{sum}$ and $\text{CS}_{lat}$. Section 5 concludes the paper.

## 2. Preliminaries

**Notation.** Given a metric space on vertices $V$, where the travel time between vertices $x_1 \in V$ and $x_2 \in V$ is denoted by $w(x_1, x_2)$, note that the travel times $w(x_1, x_2)$ for all $x_1, x_2 \in V$ are non-negative, symmetric, and satisfy the triangle inequality. Furthermore, we extended the notation of travel time between two locations to the travel time of a path: $w(x_1, x_2, \ldots, x_k) = \sum_{i=1}^{k-1} w(x_i, x_{i+1})$. In the ride-sharing problem, we are given $n$ cars, denoted by $D = \{1, 2, \ldots, n\}$, each car $k$ consisting of its location $d_k \in V$ and $m$ requests $R = \{r_1, r_2, \ldots, r_m\}$, each $r_i$ consisting of a source (pick-up location) and destination (drop-

off location) pair $(s_i, t_i) \in V \times V$. Each car can serve at most $c$ requests. We want to find an *allocation:*

$$M = \{(k, R_k) : k \in D, R_k \subseteq R, |R_k| \leq c, R_1, R_2, .., R_n \text{ are pairwise disjoint}\},$$

serving the maximum number of requests while minimizing the total travel time or minimizing the total latency. In the basic setting, we suppose $m = c \cdot n$ (see Section 3.3 for the case $m < c \cdot n$ and $m > c \cdot n$); thus, $|R_k| = c$ holds for all $k \in D$ of any feasible solution. We now elaborate on these two objectives.

**Minimizing total travel time:** For each $(k, R_k) \in M$ ($k \in D$) where $R_k$ contains $c$ requests, i.e., $|R_k| = c$, we denote the minimum travel time of serving all requests in $R_k$ by $\mathrm{cost}(k, R_k)$, i.e., the minimum time (or distance) of visiting all locations $\{s_i, t_i \mid i \in R_k\}$ starting from $d_k$ where $s_i$ is visited before $t_i$. The length of the shortest Hamiltonian path of visiting all locations $\{s_i, t_i \mid i \in R_k\}$ starting from $s_r$ ($r \in R_k$) with $s_i$ visited before $t_i$ is denoted by $\mathrm{SHP}(s_r, R_k)$. We view $\mathrm{cost}(k, R_k)$ as consisting of two parts: one term $w(d_k, s_r)$ expressing the travel time between $d_k$ and the first pick-up location $s_r$ and another term $\mathrm{SHP}(s_r, R_k)$ ($r \in R_k$) capturing the minimum travel time of visiting all requests starting from $s_r$. (It is true that, in general, the SHP is NP-hard; however, in our case, the parameter c is a small constant leading to instances of a SHP of bounded size.) The travel time needed to serve requests in $R_k$ ($k \in D$) is then given by:

$$\mathrm{cost}(k, R_k) = \min_{r \in R_k} \{w(d_k, s_r) + \mathrm{SHP}(s_r, R_k)\}. \tag{1}$$

We denote the travel time of an allocation $M$ by:

$$\mathrm{cost}(M) = \sum_{(k, R_k) \in M} \mathrm{cost}(k, R_k). \tag{2}$$

In $\mathrm{CS}_{sum}$ and $\mathrm{CS}_{sum,s=t}$, the goal is to find an allocation $M$ that minimizes $\mathrm{cost}(M)$.

**Minimizing total latency:** Here, we focus on the waiting time as perceived by an individual customer, from the moment the car leaves his/her location until the moment the customer reaches his/her drop-off location. For each $(k, R_k) \in M$ ($k \in D$) where $R_k$ contains $c$ requests, i.e., $|R_k| = c$, we denote the minimum total waiting time of all requests in $R_k$ by $\mathrm{wait}(k, R_k)$, i.e., the sum of the times to reach all drop-off locations $t_r$ ($r \in R_k$) following a path that visits all locations $\{s_i, t_i \mid i \in R_k\}$ starting from $d_k$ while $s_i$ is visited before $t_i$. We view $\mathrm{wait}(k, R_k)$ as consisting of two parts: one term $c \cdot w(d_k, s_r)$ expressing the waiting time between $d_k$ and the first pick-up location $s_r$; another term $\mathrm{SHWP}(s_r, R_k)$ capturing the sum of waiting times from the first pick-up location $s_r$ to every other drop-off location, minimized over all feasible ways of traveling through the locations in $R_k$. The latency needed to serve requests in $R_k$ ($k \in D$) is then given by:

$$\mathrm{wait}(k, R_k) = \min_{r \in R_k} \{c \cdot w(d_k, s_r) + \mathrm{SHWP}(s_r, R_k)\}. \tag{3}$$

We denote the latency of an allocation $M$ by:

$$\mathrm{wait}(M) = \sum_{(k, R_k) \in M} \mathrm{wait}(k, R_k). \tag{4}$$

Thus, in $\mathrm{CS}_{lat}$ and $\mathrm{CS}_{lat,s=t}$, the goal is to find an allocation $M$ that minimizes $\mathrm{wait}(M)$.

Another variant of the ride-sharing problem considering the latency objective is counted with respect to the pick-up location rather than the drop-off location in $\mathrm{CS}_{lat}$. In this setting, the drop-off location clearly becomes irrelevant to the objective, and our approximation results for $\mathrm{CS}_{lat,s=t}$ become valid for this variant.

### 3. The Transportation Algorithm and Its Analysis

We describe the transportation algorithm in Section 3.1 and analyze its performance for $CS_{sum}$, $CS_{sum,s=t}$, $CS_{lat}$, and $CS_{lat,s=t}$ in Section 3.2.

*3.1. The Transportation Algorithm*

In this section, we present the transportation algorithm. The idea of the algorithm is to assign to each car $k \in D$ $c$ requests based only on the travel times between the car locations $d_k$ and the request locations $s_r, t_r$, thereby ignoring travel times between different request locations.

We implemented this idea by replacing each car $k \in D$ by $c$ virtual cars $\{\gamma_1(k), \ldots, \gamma_c(k)\}$, resulting in car sets $\Gamma = \{\gamma_1(1), \ldots, \gamma_c(1), \ldots, \gamma_1(n), \ldots, \gamma_c(n)\}$ with $|\Gamma| = c \cdot n$. Next, we assigned $c \cdot n$ requests to the $c \cdot n$ cars using a particular definition of the cost $v_1(\gamma_i(k), r)$ (or $v_2(\gamma_i(k), r)$) between a request $r \in R$ and a car $\gamma_i(k) \in \Gamma$:

$$v_1(\gamma_i(k), r) = \begin{cases} w(d_k, s_r, t_r) + w(t_r, d_k) & \text{if } i < c \\ w(d_k, s_r, t_r) & \text{if } i = c \end{cases}. \tag{5}$$

$$v_2(\gamma_i(k), r) = \begin{cases} (c - i + 1)w(d_k, s_r, t_r) + (c - i)w(t_r, d_k) & \text{if } i < c \\ w(d_k, s_r, t_r) & \text{if } i = c \end{cases}. \tag{6}$$

Next, we introduce how to assign $c \cdot n$ requests to the $c \cdot n$ cars. As is showed in Algorithm 1.

---

**Algorithm 1** Transportation algorithm (TA($v$)).

---

1: **Construct a graph:** Let $G_1 \equiv (\Gamma \cup R, v_1)$ (resp. $G_2 \equiv (\Gamma \cup R, v_2)$) be the complete bipartite graph with *left* vertex-set $\Gamma$, *right* vertex-set $R$, and edge weights $v_1(\gamma_i(k), r)$ (resp. $v_2(\gamma_i(k), r)$) for $\gamma_i(k) \in \Gamma$ and $r \in R$.
2: **Find a min-weight assignment:** Find a minimum weight assignment $M_1$ (resp, $M_2$) in $G_1 \equiv (\Gamma \cup R, v_1)$ (resp, $G_2 \equiv (\Gamma \cup R, v_2)$) with weight $v_1(M_1)$ (resp. $v_2(M_2)$).
3: **Output:** $TA(v_1) \equiv \{(k, \{r_1, \ldots, r_c\}) : (\gamma_1(k), r_1), \ldots, (\gamma_c(k), r_c) \in M_1\}$.
   $TA(v_2) \equiv \{(k, \{r_1, \ldots, r_c\}) : (\gamma_1(k), r_1), \ldots, (\gamma_c(k), r_c) \in M_2\}$.

---

A solution is then found by letting car $k \in D$ serve the requests assigned to virtual cars $\{\gamma_1(k), \ldots, \gamma_c(k)\}$. Let $R_k = \{r_1, r_2, \ldots, r_c\}$ ($k \in D$) denote the requests assigned to a car $k$, where request $r_i \in R_k$ is assigned to $\gamma_i(k)$.

In our algorithm, two minimum weight assignments based on these costs are found: $M_1$ with weight $v_1(M_1)$ and $M_2$ with weight $v_2(M_2)$. We use $M_1$ to construct a solution for $CS_{sum}$ and $M_2$ to construct a solution for $CS_{lat}$.

Observe that $v_1(M_1) = \sum_{(\gamma_i(k), r) \in M_1} v_1(\gamma_i(k), r)$. This amounts to a solution where each car $k \in D$ travels according to the following path:

$$d_k \rightarrow s_{r_1} \rightarrow t_{r_1} \rightarrow d_k \rightarrow s_{r_2} \rightarrow t_{r_2} \rightarrow \cdots \rightarrow d_k \rightarrow s_{r_c} \rightarrow t_{r_c}.$$

Notice that, due to the triangle inequality, the cost of such a path will not increase by "short-cutting" the path, i.e., by traveling from each $t_{r_i}$ directly to $s_{r_{i+1}}$:

$$d_k \rightarrow s_{r_1} \rightarrow t_{r_1} \rightarrow s_{r_2} \rightarrow t_{r_2} \rightarrow \cdots \rightarrow t_{r_{c-1}} \rightarrow s_{r_c} \rightarrow t_{r_c}.$$

In fact, we use $TA(v_1)$ to denote this resulting solution found by the TA for $CS_{sum}$, with $cost(TA(v_1))$ denoting its cost.

We conclude:

$$cost(TA(v_1)) \leq v_1(M_1). \tag{7}$$

A similar observation can be made with respect to $M_2$. The quantity $v_2(M_2)$ collects the waiting time of all requests by following, for each car $k \in D$, the path:

$$d_k \rightarrow s_{r_1} \rightarrow t_{r_1} \rightarrow d_k \rightarrow s_{r_2} \rightarrow t_{r_2} \rightarrow \cdots \rightarrow d_k \rightarrow s_{r_c} \rightarrow t_{r_c}.$$

As argued above, shortcutting gives us then a feasible solution for an instance of $CS_{lat}$ we denote by $TA(v_2)$ with cost $wait(TA(v_2))$. We have:

$$wait(TA(v_2)) \leq v_2(M_2). \tag{8}$$

Recall that our problem does not force the driver to return to the original position. This implies that the cost of a driver when serving a set of request $R_k$ does not include the time from the last drop-off location to the driver's original location. This explains why in the expression for $v_1(M_1)$ (also, $v_2(M_2)$), we can subtract the corresponding travel time from the total travel time. We now give two lemmas concerning $v_1(M_1)$ (which we need to prove Theorem 1) and two more lemmas concerning $v_2(M_2)$ (which we need to prove Theorem 2).

**Lemma 1.** *For any $c \geq 2$, we have:*

$$v_1(M_1) = \sum_{(k,R_k) \in TA(v_1)} \left( \sum_{r \in R_k} w(d_k, s_r, t_r, d_k) - \max_{r \in R_k} w(d_k, t_r) \right).$$

**Proof.** We claim that $v_1(M_1)$ is minimized if and only if, for each car $k \in D$ and $(\gamma_c(k), r_c) \in M_1$, $r_c = \arg\max_{r \in R_k} w(d_k, t_r)$. If this claim holds, then based on the definition of the cost $v_1(\cdot, \cdot)$, we have $v_1(k, R_k) = \sum_{r \in R_k} w(d_k, s_r, t_r, d_k) - w(d_k, t_{r_c}) = \sum_{r \in R_k} w(d_k, s_r, t_r, d_k) - \max_{r \in R_k} w(d_k, t_r)$, and thus:

$$v_1(M_1) = \sum_{(k,R_k) \in TA(v_1)} \left( \sum_{r \in R_k} w(d_k, s_r, t_r, d_k) - \max_{r \in R_k} w(d_k, t_r) \right).$$

It remains to prove the claim. Consider any $R_k = \{r_1, r_2, \ldots, r_c\}$ for car $k \in D$. We prove that $v_1(M_1)$ is minimized if and only if $w(d_k, t_{r_c}) \geq w(d_k, t_{r_x})$ for all $r_x \in R_k$.

*Necessary condition:* Since $v_1(k, R_k)$ is minimized, we have $w(d_k, s_{r_x}, t_{r_x}) + w(d_k, t_{r_x}) + w(d_k, s_{r_c}, t_{r_c}) \leq w(d_k, s_{r_c}, t_{r_c}) + w(d_k, t_{r_c}) + w(d_k, s_{r_x}, t_{r_x})$ based on the definition of $v_1$ (see Equation (5)), then $w(d_k, t_{r_x}) \leq w(d_k, t_{r_c})$ holds.

*Sufficient condition:* Since $w(d_k, t_{r_c}) \geq w(d_k, t_{r_x})$, then $w(d_k, s_{r_c}, t_{r_c}) + w(d_k, t_{r_c}) + w(d_k, s_{r_x}, t_{r_x}) \geq w(d_k, s_{r_x}, t_{r_x}) + w(d_k, t_{r_x}) + w(d_k, s_{r_c}, t_{r_c})$, and that means $v_1(M_1)$ is minimized as $r_c = \arg\max_{r \in R_k} w(d_k, t_r)$.  □

From the above lemma and the fact that $M_1$ is a minimum weight assignment in $G_1 \equiv (\Gamma \cup R, v_1)$, we have the following lemma:

**Lemma 2.** *For $c \geq 2$ and for each allocation M, we have:*

$$v_1(M_1) \leq \sum_{(k,R_k) \in M} \left( \sum_{r \in R_k} w(d_k, s_r, t_r, d_k) - \max_{r \in R_k} w(d_k, t_r) \right).$$

We now provide two lemmas concerning $v_2(M_2)$. In the statement of these lemmas, we index the requests such that, for each $k \in D$, $R_k = \{r_1, r_2, \ldots, r_c\}$.

**Lemma 3.** *For any $c \geq 2$, for each car $k \in D$ and $\forall r_x, r_y \in R_k$ with $x < y$, $w(d_k, s_{r_x}, t_{r_x}) + w(d_k, t_{r_x}) \leq w(d_k, s_{r_y}, t_{r_y}) + w(d_k, t_{r_y})$.*

**Proof.** We claim that $v_2(M_2)$ is minimized if and only if for each car $k \in D$ and $\forall r_x, r_y \in R_k = \{r_1, r_2, \ldots, r_c\}$ with $x < y$, $w(d_k, s_{r_x}, t_{r_x}) + w(d_k, t_{r_x}) \le w(d_k, s_{r_y}, t_{r_y}) + w(d_k, t_{r_y})$. Consider $\forall r_x, r_y \in R_k = \{r_1, r_2, \ldots, r_c\}$ with $x < y$ for car $k \in D$. We prove that $v_2(M_2)$ is minimized if and only if $w(d_k, s_{r_x}, t_{r_x}) + w(d_k, t_{r_x}) \le w(d_k, s_{r_y}, t_{r_y}) + w(d_k, t_{r_y})$.

*Necessary condition:* Since $v_2(M_2)$ is minimized, based on definition of cost $v_2(\cdot, \cdot)$, we have:

$$(c - x + 1)w(d_k, s_{r_x}, t_{r_x}) + (c - x)w(t_{r_x}, d_k) + (c - y + 1)w(d_k, s_{r_y}, t_{r_y}) + (c - y)w(t_{r_y}, d_k)$$
$$\le (c - x + 1)w(d_k, s_{r_y}, t_{r_y}) + (c - x)w(t_{r_y}, d_k) + (c - y + 1)w(d_k, s_{r_x}, t_{r_x}) + (c - y)w(t_{r_x}, d_k).$$

$$\iff (y - x)(w(d_k, s_{r_x}, t_{r_x}) + w(d_k, t_{r_x})) \le (y - x)(w(d_k, s_{r_y}, t_{r_y}) + w(d_k, t_{r_y}))$$

Thus, $w(d_k, s_{r_x}, t_{r_x}) + w(d_k, t_{r_x}) \le w(d_k, s_{r_y}, t_{r_y}) + w(d_k, t_{r_y})$, since $x < y$.

*Sufficient condition:* According to the above statement, the condition $w(d_k, s_{r_x}, t_{r_x}) + w(d_k, t_{r_x}) \le w(d_k, s_{r_y}, t_{r_y}) + w(d_k, t_{r_y})$ implies that $v_2(M_2)$ is minimized. □

Since $M_2$ is a minimum weight assignment in $G_2 \equiv (\Gamma \cup R, v_2)$, we have the following lemma:

**Lemma 4.** *For $c \ge 2$ and for each allocation $M$, we have:*

$$v_2(M_2) \le \sum_{(k, R_k) \in M} \sum_{i=1}^{c} ((c - i + 1) \cdot w(d_k, s_{r_i}, t_{r_i}) + (c - i) \cdot w(d_k, t_{r_i})).$$

*3.2. Approximation Analysis of the TA*

Let us denote an optimal allocation in $\mathrm{CS}_{sum}$ by $M^* = \{(k, R_k^*) : k \in D\}$. Let $M_R^* = \{R_k^* : (k, R_k^*) \in M^*\}$ denote the collection of $c$-tuples of requests in an optimal solution $M^*$. We now establish the worst-case ratios of the $\mathrm{TA}(v_1)$ for $\mathrm{CS}_{sum}$ and $\mathrm{CS}_{sum,s=t}$.

**Theorem 1.** *The $\mathrm{TA}(v_1)$ is a $(2c - 1)$-approximation algorithm for $\mathrm{CS}_{sum}$. Moreover, there exists an instance $I$ of $\mathrm{CS}_{sum,s=t}$ for which $\mathrm{cost}(\mathrm{TA}(v_1)(I)) = (2c - 1) \cdot \mathrm{cost}(M^*(I))$.*

**Proof.**

$$\mathrm{cost}(\mathrm{TA}(v_1)) \le v_1(M_1) \tag{9}$$

$$\le \sum_{(k, R_k^*) \in M^*} \left( \sum_{r \in R_k^*} w(d_k, s_r, t_r, d_k) - \max_{r \in R_k^*} w(d_k, t_r) \right) \tag{10}$$

$$= \sum_{(k, R_k^*) \in M^*} \left( \sum_{r \in R_k^*} w(d_k, s_r, t_r) + \sum_{r \in R_k^*} w(t_r, d_k) - \max_{r \in R_k^*} w(d_k, t_r) \right) \tag{11}$$

$$\le \sum_{(k, R_k^*) \in M^*} (2c - 1)\, \mathrm{cost}(k, R_k^*) \tag{12}$$

$$= (2c - 1)\, \mathrm{cost}(M^*) \tag{13}$$

We now comment on the validity of the inequalities above. Inequality (9) follows from applying Inequality (7), and Inequality (10) follows from Lemma 2. The final Inequality (12) follows from the fact that for any $r \in R_k^*$, $w(d_k, t_r) \le w(d_k, s_r, t_r) \le \mathrm{cost}(k, R_k^*)$.

To see that the bound $2c - 1$ is tight even for $\mathrm{CS}_{sum,s=t}$, consider the instance $I$ depicted in Figure 1. This instance has $c$ cars $D = \{k_1, k_2, \ldots, k_c\}$ with car locations $\{d_1, d_2, \ldots, d_c\}$ and $c^2$ requests $R = \{1, 2, \ldots, c^2\}$ with locations $\{s_1, s_2, \ldots, s_{c^2}\}$ (the pick-up and drop-off locations are identical for each request). Locations corresponding to distinct vertices in Figure 1 are at Travel Time 1. Observe that an optimal solution is $M^*(I) = \{(k_1, \{1, 2, \ldots, c\}), (k_2, \{c + 1, c + 2, \ldots, 2c\}), \ldots, (k_c, \{c(c - 1) + 1, c(c - 1) + 2, \ldots, c^2\})\}$ with $\mathrm{cost}(M^*(I)) = c$.

Let us now analyze the performance of $\mathrm{TA}(v_1)$ on instance $I$. Notice that $\mathrm{TA}(v_1)$ may assign requests $\{i, c + i, \ldots, (c - 1)c + i\}$ to car $k_i$. In that case, the total cost of $\mathrm{TA}(v_1)$ is $c(2(c - 1) + 1)) = c(2c - 1)$, showing tightness. $\quad\square$

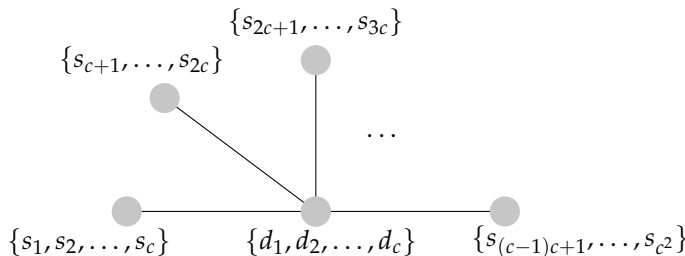

**Figure 1.** A worst-case instance for the transportation algorithm.

We proceed by establishing the worst-case ratios of the $\mathrm{TA}(v_2)$ for $\mathrm{CS}_{lat}$ and $\mathrm{CS}_{lat,s=t}$. Again, we assume that an optimal solution to $\mathrm{CS}_{lat}$ is denoted by $M^*$, and the collection of $c$-tuples of requests in $M^*$ is denoted by $M_R^* = \{R_k^* : (k, R_k^*) \in M^*\}$. In the following theorem, we index the requests such that, for each $k \in D$, $R_k^* = \{r_1, r_2, \ldots, r_c\}$.

**Theorem 2.** *The* $\mathrm{TA}(v_2)$ *is a c-approximation algorithm for* $\mathrm{CS}_{lat}$. *Moreover, there exists an instance I of* $\mathrm{CS}_{lat,s=t}$ *for which* $\mathrm{wait}(\mathrm{TA}(v_2)(I)) = c \cdot \mathrm{wait}(M^*(I))$.

**Proof.**

$$\mathrm{wait}(\mathrm{TA}(v_1)) \leq v_2(M_2) \tag{14}$$

$$\leq \sum_{(k,R_k)\in M^*} \sum_{i=1}^{c} ((c - i + 1) \cdot w(d_k, s_{r_i}, t_{r_i}) + (c - i) \cdot w(d_k, t_{r_i})) \tag{15}$$

$$= \sum_{(k,R_k^*)\in M^*} \sum_{i=1}^{c} (c \cdot w(d_k, s_{r_i}, t_{r_i}) - (i - 1) \cdot w(d_k, s_{r_i}, t_{r_i}) + (c - i) \cdot w(d_k, t_{r_i})) \tag{16}$$

$$\leq \sum_{(k,R_k^*)\in M^*} \sum_{i=1}^{c} c \cdot w(d_k, s_{r_i}, t_{r_i}) \tag{17}$$

$$\leq \sum_{(k,R_k^*)\in M^*} c \cdot \mathrm{wait}(k, R_k^*) \tag{18}$$

$$= c \cdot \mathrm{wait}(M^*) \tag{19}$$

We now comment on the validity of the inequalities above. Inequality (14) follows from applying Inequality (8), and Inequality (15) follows from Lemma 4. Inequality (17) follows from (we prove it later):

$$\sum_{i=1}^{c} (-(i - 1) \cdot w(d_k, s_{r_i}, t_{r_i}) + (c - i) \cdot w(d_k, t_{r_i})) \leq 0.$$

The final Inequality (18) follows from the fact that $\sum_{i=1}^{c} w(d_k, s_r, t_r) \leq \mathrm{wait}(k, R_k^*)$. Notice that:

$$\sum_{i=1}^{c} (-(i - 1) \cdot w(d_k, s_{r_i}, t_{r_i}) + (c - i) \cdot w(d_k, t_{r_i}))$$

$$\leq \sum_{i<\frac{c+1}{2}} (c - 2i + 1) \cdot w(d_k, t_{r_i}) - \sum_{i\geq\frac{c+1}{2}} (2i - c - 1) \cdot w(d_k, s_{r_i}, t_{r_i})$$

$$= \sum_{i<\frac{c+1}{2}} ((c - 2i + 1) \cdot w(d_k, t_{r_i}) - (2(c + 1 - i) - c - 1) \cdot w(d_k, s_{r_{c+1-i}}, t_{r_{c+1-i}}))$$

$$\leq 0$$

where the first inequality follows from the triangle inequality; the second inequality follows from $w(d_k, t_{r_x}) \leq w(d_k, s_{r_y}, t_{r_y})$ for all $r_x, r_y \in R_k^*$ with $x < y$ since $w(d_k, s_{r_x}, t_{r_x}) + w(d_k, t_{r_x}) \leq w(d_k, s_{r_y}, t_{r_y}) + w(d_k, t_{r_y})$ by Lemma 3.

To see that the bound $c$ is tight even for $CS_{lat,s=t}$, consider the instance depicted in Figure 1. Observe that an optimal solution is $M^*(I) = \{(k_1, \{1, 2, \ldots, c\}), (k_2, \{c + 1, c + 2, \ldots, 2c\}), \ldots, (k_c, \{c(c - 1) + 1, c(c - 1) + 2, \ldots, c^2\})\}$ with $\text{wait}(M^*(I)) = c^2$. Let us now analyze the performance of $TA(v_2)$ on instance $I$. $TA(v_2)$ may assign requests $\{i, c + i, \ldots, (c - 1)c + i\}$ to car $k_i$. In that case, the total waiting time of $TA(v_2)$ is $c \cdot (1 + 3 + \cdots + (2(i - 1) + 1) + \cdots + 2(c - 1) + 1)) = c(c \cdot (1 + 2c - 1)/2) = c^3$, showing tightness. □

*3.3. Discussion*

Clearly, the TA is a polynomial-time algorithm, and it is easy to implement; moreover, it can be generalized to handle a variety of situations. We now list three situations and briefly comment on the corresponding worst-case behavior:

- **Ride-sharing with car-dependent speeds or *related* ride-sharing.** In this situation, the cars have speed $p_1, p_2, \ldots, p_n$. The travel time of serving requests in $R_k$ is denoted by $\text{cost}(k, R_k)/p_k$, and the total travel time of an allocation $M$ is denoted by $\sum_{(k, R_k) \in M} \text{cost}(k, R_k)/p_k$. Analogously, the total latency of an allocation $M$ is denoted by $\sum_{(k, R_k) \in M} \text{wait}(k, R_k)/p_k$. Without going into the details, we point out that the TA can be modified by appropriately redefining $v_1(k, r)$ and $v_2(k, r)$ in terms of the cost above; we claim that the corresponding worst-case ratios of TA as shown in Table 1 remain unchanged;

- **Car redundancy: $c \cdot n > m$.** In this situation, our problem is to find an allocation that serves all requests with the minimum total cost (total travel time or total latency). Clearly, some cars may serve less than $c$ requests, or even do not serve a request. To apply TA for this situation, we need to add a number of requests to fill the shortage of requests, without affecting the total travel time or latency. We created an instance of our problem by adding a number of dummy requests $R_d$ with $|R_d| = c \cdot n - m$, where the travel time between a request in $R_d$ and a car in $D$ is zero, i.e., $v_1(\gamma_i(k), r) = 0$ and $v_2(\gamma_i(k), r) = 0$ for all $i \in [c], k \in D, r \in R_d$. Since the cost of assigning dummy requests is zero in any feasible solution, removing dummy requests of a solution for the newly created instance with $c \cdot n = |R|$ will give us a solution to the original instance;

- **Car deficiency: $c \cdot n < m$.** In this situation, our problem is to find an allocation that serves the maximum number of requests ($c \cdot n$ requests) with the minimum total cost (total travel time or total latency). It follows that some requests will not be served. To apply the TA for this situation, we created an instance of our problem by adding a number of dummy cars $D_d$ with $|D_d| = m - n \cdot c$, where the travel time between a car in $D_d$ and a request in $R$ is $H$ ($H$ is a sufficiently large number), i.e., $v_1(k, r) = H$ and $v_2(k, r) = H$ for all $k \in D_d$, $r \in R$. Removing dummy cars (and their corresponding requests) gives us a solution to the original instance. Since we found an assignment with the minimum total weight and we removed the set of requests assigned to the dummy cars, we claim that the TA selected $c \cdot n$ requests with the minimum total weight. In fact, the proofs in Section 3.1 imply that the $TA(v_1)$ is a $(2c - 1)$-approximation algorithm for $CS_{sum}$ and the $TA(v_2)$ is a $c$-approximation algorithm for $CS_{lat}$.

## 4. The Case $c = 2$: Algorithms and Their Analysis

In this section, we consider the ride-sharing problems $CS_{sum}$, $CS_{sum,s=t}$, $CS_{lat}$, and $CS_{lat,s=t}$ with capacity $c = 2$ and each car serving exactly two requests, i.e., $m = 2n$ (see [22]). In Section 4.1, we propose and analyze the match-and-assign algorithm (MA). Next, in Section 4.2, we analyze the combined algorithm (CA), i.e., the better of the two algorithms, the MA and TA.

For convenience, we explicitly write the quantity $SHP(s_i, \{i, j\})$ in $CS_{sum}$ by a parameter $u_{ij}$ as follows:

$$
\begin{aligned}
u_{ij} \equiv\ & \min\{w(s_i, s_j, t_i, t_j), w(s_i, s_j, t_j, t_i), w(s_i, t_i, s_j, t_j)\} \\
& \text{for each } i, j \in R \times R, i \neq j.
\end{aligned}
\tag{20}
$$

Notice that the $u_{ij}$'s are not necessarily symmetric. Obviously, $u_{ij} \geq w(s_i, s_j)$ and $u_{ji} \geq w(s_i, s_j)$. For $CS_{sum,s=t}$, we have $u_{ij} = u_{ji} \equiv w(s_i, s_j)$.

The travel time needed to serve requests in $R_k = \{i, j\}$ ($k \in D$) is then given by:

$$
\text{cost}(k, \{i, j\}) = \min\{w(d_k, s_i) + u_{ij}, w(d_k, s_j) + u_{ji}\}.
\tag{21}
$$

For convenience, we also explicitly write the quantity $SHWP(s_i, \{i, j\})$ in $CS_{sum}$ by a parameter $\mu_{ij}$ as follows:

$$
\begin{aligned}
\mu_{ij} \equiv\ & \min\{w(s_i, s_j, t_i) + w(s_i, s_j, t_i, t_j), w(s_i, s_j, t_j) + w(s_i, s_j, t_j, t_i), \\
& w(s_i, t_i) + w(s_i, t_i, s_j, t_j)\} \text{ for each } i, j \in R \times R, i \neq j.
\end{aligned}
\tag{22}
$$

Notice that the $\mu_{ij}$'s are not necessarily symmetric. For $CS_{lat,s=t}$, we have $\mu_{ij} = \mu_{ji} \equiv w(s_i, s_j)$.

The latency needed to serve requests in $R_k = \{i, j\}$ ($k \in D$) is then given by:

$$
\text{wait}(k, \{i, j\}) = \min\{2w(d_k, s_i) + \mu_{ij}, 2w(d_k, s_j) + \mu_{ji}\}.
\tag{23}
$$

### 4.1. The Match-and-Assign Algorithm and Its Analysis

We came up with a match-and-assign algorithm, the $MA(\alpha, v)$, the idea being that, first, requests are matched into request pairs, after which the request pairs are assigned to the cars. Finding request pairs is performed by using a carefully chosen time $v_3(\{i, j\})$ between a pair of requests $\{i, j\}$, as well as a travel time $v_4(k, \{i, j\})$ between each request pair $\{i, j\}$ and a car $k \in D$:

$$
v_3(\{i, j\}) \equiv \frac{v_{ij} + v_{ji}}{2}, \quad v \in \{u, \mu\}.
\tag{24}
$$

$$
v_4(k, \{i, j\}) \equiv \min\{\alpha w(d_k, s_i) + \frac{v_{ij} - v_{ji}}{2}, \alpha w(d_k, s_j) - \frac{v_{ij} - v_{ji}}{2}\}, \quad \alpha \in \{1, 2\}, v \in \{u, \mu\}.
\tag{25}
$$

Now, we introduce the match-and-assign Algorithm 2.

The resulting quantity is $v_3(M_3) + v_4(M_4)$; we now prove two lemmas concerning this quantity, which will be of use in the approximation analysis.

---

**Algorithm 2** Match-and-assign algorithm (MA($\alpha, v$)).

---

1: **Matching step:**
   - **Construct a graph:** Let $G_3 \equiv (R, v_3)$ be the complete weighted graph where an edge between vertex $i \in R$ and vertex $j \in R$ has weight $v_3(\{i, j\})$;
   - **Find a min-weight matching:** Find a minimum weight perfect matching $M_3$ in $G_3 \equiv (R, v_3)$ with weight $v_3(M_3)$.
2: **Assignment step:**
   - **Construct a graph:** Let $G_4 \equiv (D \cup M_3, v_4)$ be the complete bipartite graph with *left* vertex-set $D$, *right* vertex-set $M_3$, and edges with weight $v_4(k, \{i, j\})$ for $k \in D$, and $\{i, j\} \in M_3$;
   - **Find a min-weight assignment:** Find a minimum weight assignment $M_4$ in $G_4 \equiv (D \cup M_3, v_4)$ with weight $v_4(M_4)$.
3: **Output:** MA $= M_4$.

---

**Lemma 5.** *For each $\alpha \in \{1, 2\}$ and $v \in \{u, \mu\}$, we have:*

$$v_3(M_3) + v_4(M_4) = \sum_{(k, \{i,j\}) \in M_4} \min\{\alpha w(d_k, s_i) + v_{ij}, \alpha w(d_k, s_j) + v_{ji}\}.$$

**Proof.** Without loss of generality, for any $\{i, j\} \in M_3$, suppose $v_{ij} - v_{ji} \geq 0$ (the other case is symmetric).

$$v_3(M_3) + v_4(M_4) = \sum_{\{i,j\} \in M_3} \frac{v_{ij} + v_{ji}}{2} +$$

$$= \sum_{(k, \{i,j\}) \in M_4} \min\{\alpha w(d_k, s_i) + \frac{v_{ij} - v_{ji}}{2}, \alpha w(d_k, s_j) - \frac{v_{ij} - v_{ji}}{2}\}$$

$$= \sum_{(k, \{i,j\}) \in M_4} \min\{\alpha w(d_k, s_i) + v_{ij}, \alpha w(d_k, s_j) + v_{ji}\}.$$

The first equality follows from the definition of $v_3$ and $v_4$ (see Equations (24) and (25)). □

**Lemma 6.** *For $\alpha \in \{1, 2\}$, $v \in \{u, \mu\}$, and for each allocation $M$, we have:*

$$v_3(M_3) + v_4(M_2) \leq \sum_{(k, \{i,j\}) \in M} \frac{\alpha w(d_k, s_i) + \alpha w(d_k, s_j) + v_{ij} + v_{ji}}{2}.$$

**Proof.** For an allocation $M$, let $M_R = \{R_k : (k, R_k) \in M\}$. Observe that:

$$v_3(M_3) \leq \sum_{\{i,j\} \in M_R} \frac{v_{ij} + v_{ji}}{2}, \tag{26}$$

since $M_3$ is a minimum weight perfect matching in $G_3 \equiv (R, v_3)$.

We claim that:

$$v_4(M_4) \leq \sum_{(k, \{i,j\}) \in M} \frac{\alpha w(d_k, s_i) + \alpha w(d_k, s_j)}{2}. \tag{27}$$

When summing (26) and (27), the lemma follows.

Hence, it remains to prove (27). Consider an allocation $M$, and consider the matching $M_3$ found in the first step of the MA. Based on $M$ and $M_3$, we construct the graph $G' = (R \cup D, M_1 \cup \{(\{i, k\}, \{j, k\}) : (k, \{i, j\}) \in M\})$. Note that every vertex in graph $G'$ has degree two. Thus, we can partition $G'$ into a set of disjoint cycles called $C$; each cycle $c \in C$

can be written as $c = (i_1, j_1, k_1, i_2, j_2, k_2, \ldots, k_h, i_1)$, where $\{i_s, j_s\} \in M_3$, $(k_s, \{j_s, i_{s+1}\}) \in M$ for $1 \leq s < h$ and $(k_h, \{j_h, i_1\}) \in M$. Consider now, for each cycle $c \in C$, the following two assignments called $M_\ell^c$ and $M_r^c$:

- $M_\ell^c = \{(\{i_1, j_1\}, k_1), (\{i_2, j_2\}, k_2), \ldots, (\{i_h, j_h\}, k_h)\}$,
- $M_r^c = \{(k_1, \{i_2, j_2\}), (k_2, \{i_3, j_3\}), \ldots, (k_h, \{i_1, j_1\})\}$.

Obviously, both $M_\ell \equiv \bigcup_{c \in C} M_\ell^c$ and $M_r \equiv \bigcup_{c \in C} M_r^c$ are a feasible assignment in $G_4 = (D \cup M_3, v_4)$. Given the definition of $v_4(k, \{i, j\})$ (see Equation (25)), we derive for each pair of requests $\{i, j\}$ and two cars $a, b$: $v_4(a, \{i, j\}) + v_2(b, \{i, j\}) \leq \alpha w(d_a, s_i) + \frac{v_{ij} - v_{ji}}{2} + \alpha w(d_b, s_j) - \frac{v_{ij} - v_{ji}}{2} = \alpha(w(d_a, s_i) + w(d_b, s_j))$. Similarly, it follows that: $v_4(a, \{i, j\}) + v_4(b, \{i, j\}) \leq \alpha(w(d_a, s_j) + w(d_b, s_i))$. Thus, for each $c \in C$:

$$\sum_{(k, \{i,j\}) \in M_\ell^c} v_4(k, \{i, j\}) + \sum_{(k, \{i,j\}) \in M_r^c} v_4(k, \{i, j\}) \leq \sum_{\substack{\{i,k\}, \{j,k\} \in c \\ (k, \{i,j\}) \in M}} \alpha(w(d_k, s_i) + w(d_k, s_j)). \quad (28)$$

Note that $M_4$ is a minimum weight assignment in $G_4 = (D \cup M_3, v_4)$, and both $M_\ell$ and $M_r$ are a feasible assignment in $G_4 = (D \cup M_3, v_4)$. Thus:

$$v_4(M_4) \leq \sum_{c \in C} \min\{v_4(M_\ell^c), v_4(M_r^c)\}$$

$$\leq \frac{1}{2} \sum_{c \in C} \left( \sum_{(k, \{i,j\}) \in M_\ell^c} v_4(k, \{i, j\}) + \sum_{(k, \{i,j\}) \in M_r^c} v_4(k, \{i, j\}) \right)$$

$$\leq \sum_{(k, \{i,j\}) \in M} \frac{\alpha w(d_k, s_i) + \alpha w(d_k, s_j)}{2}.$$

The last inequality follows from (28), and hence, (27) is proven. □

**Lemma 7.** *For any two requests i and j, we have:*

$$\max\{u_{ij}, u_{ji}\} - \min\{u_{ij}, u_{ji}\} \leq w(s_i, s_j);$$

*and:*

$$u_{ij} \leq 2u_{ji}.$$

Without loss of generality, suppose $u_{ij} \geq u_{ji}$, $u_{ij} \leq w(s_i, s_j) + u_{ji}$. Since $w(s_i, s_j) \leq \min\{u_{ij}, u_{ji}\}$, we have $u_{ij} \leq 2u_{ji}$ for any two requests $i$ and $j$.

**Theorem 3.** *The* $\mathrm{MA}(1, u)$ *is a two-approximation algorithm for* $\mathrm{CS}_{sum}$. *Moreover, there exists an instance I for which* $\mathrm{cost}(\mathrm{MA}(I)) = 2\mathrm{cost}(M^*(I))$.

**Proof.** We assume w.l.o.g. that, for each $(k, \{i, j\}) \in M^*$, $\mathrm{cost}(k, \{i, j\}) = w(d_k, s_i) + u_{ij}$. We have:

$$\text{cost}(\text{MA}(1,u)) = \sum_{(k,\{i,j\})\in\text{MA}} \min\{w(d_k,s_i)+u_{ij}, w(d_k,s_j)+u_{ji}\} \tag{29}$$

$$= v_3(M_3) + v_4(M_4) \tag{30}$$

$$\leq \sum_{(k,\{i,j\})\in M^*} \frac{w(d_k,s_i)+w(d_k,s_j)+u_{ij}+u_{ji}}{2} \tag{31}$$

$$\leq \sum_{(k,\{i,j\})\in M^*} \frac{2w(d_k,s_i)+w(s_i,s_j)+3u_{ij}}{2} \tag{32}$$

$$\leq \frac{1}{2} \sum_{(k,\{i,j\})\in M^*} (2w(d_k,s_i)+4u_{ij}) \tag{33}$$

$$\leq \frac{1}{2} \sum_{(k,\{i,j\})\in M^*} 4\text{cost}(k,\{i,j\}) \tag{34}$$

$$= 2\,\text{cost}(M^*). \tag{35}$$

Equation (29) follows from (2) and (21). Equation (30) follows from Lemma 5. Inequality (31) follows from Lemma 6. Inequality (32) follows from the triangle inequality, and $u_{ji} \leq 2u_{ij}$ for each request pair $\{i,j\} \in R^2$ based on Lemma 7. Inequality (33) follows from $w(s_i,s_j) \leq u_{ij}$. Notice that $\text{cost}(\text{MA}(I)) \leq 2\text{cost}(M^*(I))$ is actually tight by the instance depicted in Figure 2. □

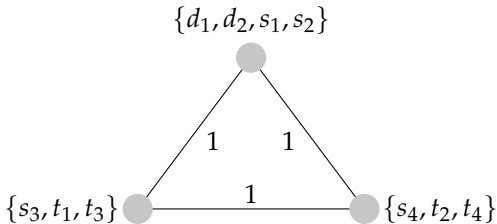

**Figure 2.** A worst-case instance for the $\text{CA}_{sum}$ of $\text{CS}_{sum}$.

**Theorem 4.** *The* $\text{MA}(1,u)$ *is a* $3/2$-*approximation algorithm for* $\text{CS}_{sum,s=t}$. *Moreover, there exists an instance I for which* $\text{cost}(\text{MA}(I)) = 3/2\,\text{cost}(M^*(I))$.

**Proof.**

$$\text{cost}(\text{MA}(1,u)) = \sum_{(k,\{i,j\})\in\text{MA}} \min\{w(d_k,s_i)+u_{ij}, w(d_k,s_j)+u_{ji}\} \tag{36}$$

$$\leq \sum_{(k,\{i,j\})\in M^*} w(s_i,s_j) + \frac{w(d_k,s_i)+w(d_k,s_j)}{2} \tag{37}$$

$$\leq \sum_{(k,\{i,j\})\in M^*} 3/2\,\text{cost}(k,\{i,j\}) \tag{38}$$

$$\leq 3/2\,\text{cost}(M^*) \tag{39}$$

Equation (36) follows from (20). Inequality (37) follows from Lemma 5 and 6, and $u_{ij} = u_{ji} = w(s_i,s_j)$.

To see that the equality may hold in $\text{cost}(\text{MA}(I)) \leq 3/2\,\text{cost}(M^*(I))$, consider the subgraph induced by the nodes $(d_1,s_3)$, $(s_1,s_2)$, and $(d_2,s_4)$ in Figure 3 with cars $\{k_1,k_2\}$ and requests $\{1,2,3,4\}$. Observe that an optimal solution is $\{(k_1,\{1,3\}),(k_2,\{2,4\})\}$ with $\text{cost}(M^*(I)) = 2$. Note that $\text{MR}^* = \{\{1,3\},\{2,4\}\}$. Let us now analyze the performance of $\text{MA}(1,u)$ on this instance. Based on the $u$ values as defined in (20), $\text{MA}(1,u)$ can find, in the first step, matching $M_3 = \{\{1,2\},\{3,4\}\}$ because $v_3(M_3) = v_3(\text{MR}^*) = 2$. Then, no matter how the second step assigns the request pairs to cars, the total waiting time of $\text{MA}(1,u)$ will be three. □

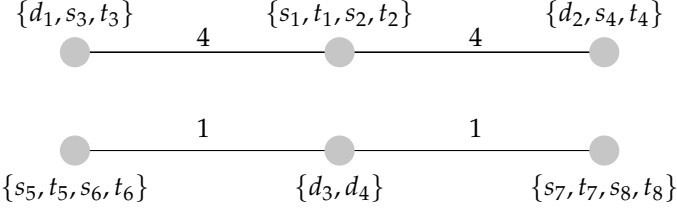

**Figure 3.** A worst-case instance for the $\text{CA}_{sum}$ of $\text{CS}_{sum,s=t}$.

**Theorem 5.** *The* $\text{MA}(2, \mu)$ *is a two-approximation algorithm for* $\text{CS}_{lat}$. *Moreover, there exists an instance I of* $\text{CS}_{lat,s=t}$ *for which* $\text{wait}(\text{MA}(I)) = 2\text{wait}(M^*(I))$.

**Proof.**

$$\text{wait}(\text{MA}(2, \mu)) = \sum_{(k,\{i,j\}) \in \text{MA}} \min\{2w(d_k, s_i) + \mu_{ij}, 2w(s_k, s_j) + \mu_{ji}\} \quad (40)$$

$$\leq \sum_{(k,\{i,j\}) \in M^*} \frac{2w(d_k, s_i) + \mu_{ij} + 2w(d_k, s_j) + \mu_{ji}}{2} \quad (41)$$

$$\leq \sum_{(k,\{i,j\}) \in M^*} \frac{4\min\{w(d_k, s_i), w(d_k, s_j)\} + \mu_{ij} + 2w(s_i, s_j) + \mu_{ji}}{2} \quad (42)$$

$$\leq \sum_{(k,\{i,j\}) \in M^*} \min\{4w(d_k, s_i) + 2\mu_{ij}, 4w(d_k, s_j) + 2\mu_{ji}\} \quad (43)$$

$$= \sum_{(k,\{i,j\}) \in M^*} 2\,\text{wait}(k, \{i, j\}) \quad (44)$$

$$= 2\,\text{wait}(M^*) \quad (45)$$

Inequality (41) follows from Lemmas 5 and 6. Inequality (42) follows from the triangle inequality. Inequality (43) follows from $w(s_i, s_j) \leq \min\{\mu_{ij}, \mu_{ji}\}$.

To see that the equality may hold in $\text{wait}(\text{MA}(I)) \leq 2\text{wait}(M^*(I))$, consider the instance $I$ depicted in Figure 4. This instance has two cars $D = \{k_1, k_2\}$ with car locations $\{d_1, d_2\}$ and four requests $R = \{1, 2, 3, 4\}$. If two points are not connected by an edge, their travel time equals five. Observe that an optimal solution is $\{(k_1, \{1,3\}), (k_2, \{2,4\})\}$ with $\text{wait}(M^*(I)) = 4$. Note that $M_R^* = \{\{1,3\}, \{2,4\}\}$. Let us now analyze the performance of the $\text{MA}(2, \mu)$ on instance $I$.

$\{d_1, s_3, t_3\}$　　　　$\{s_1, t_1, s_2, t_2\}$　　　　$\{d_2, s_4, t_4\}$

○———— 2 ————○———— 2 ————○

**Figure 4.** A worst-case instance of the $\text{MA}(2, \mu)$ for $CS_{lat,s=t}$.

Based on the $\mu$ values as defined in (22), the $\text{MA}(2, \mu)$ can find, in the first step, matching $M_3 = \{\{1,2\}, \{3,4\}\}$ because $v_3(M_3) = v_3(M_R^*) = 8$. Then, no matter how the second step assigns the request pairs to cars, the total waiting time of the $\text{MA}(2, \mu)$ will be eight. □

*4.2. The Combined Algorithm and Its Analysis*

The $\text{CA}_{sum}$ runs the $\text{MA}(1, u)$ and $\text{TA}(v_1)$ and then outputs the better of the two solutions. We now state the main result for $\text{CS}_{sum}$.

**Theorem 6.** *The* $\text{CA}_{sum}$ *is a two-approximation algorithm for* $\text{CS}_{sum}$. *Moreover, there exists an instance I for which* $\text{cost}(\text{CA}_{sum}(I)) = 2\text{cost}(M^*(I))$.

**Proof.** It is obvious that, as $\text{cost}(\text{CA}_{sum}) = \min\{\text{cost}(\text{MA}(1, u)), \text{cost}(\text{TA}(v_1))\}$, Theorems 1 and 3 imply that the $\text{CA}_{sum}$ is a two-approximation algorithm for $\text{CS}_{sum}$. We now provide an instance for which this ratio is achieved.

Consider the instance $I$ depicted in Figure 2. This instance has two cars $D = \{k_1, k_2\}$ with car locations $\{d_1, d_2\}$ and four requests $R = \{1, 2, 3, 4\}$. Locations corresponding to distinct vertices in Figure 2 are at Travel Time 1. Observe that an optimal solution is $M^*(I) = \{(k_1, \{1, 3\}), (k_2, \{2, 4\})\}$ with $\text{cost}(M^*(I)) = 2$. Note that $M_R^* = \{\{1, 3\}, \{2, 4\}\}$. Let us now analyze the performance of the $\text{MA}(1, u)$ and $\text{TA}(v_1)$ on instance $I$.

Based on the $u_{ij}$ values as defined in (20), the $\text{MA}(1, u)$ can find, in the first step, matching $M_3 = \{\{1, 2\}, \{3, 4\}\}$ with $v_3(M_3) = v_3(M_R^*) = 3$. Then, no matter how the second step assigns the request pairs to cars (since two cars stay at the same location), the total cost of $\text{MA}(1, u)$ will be four.

The $\text{TA}(v_1)$ may assign Requests 1 and 2 to Car 1 and Requests 3 and 4 to Car 2 since:

$$v_1(\{(k_1, 1), (k_1, 2), (k_2, 3), (k_2, 4)\}) = v_1(\{(k_1, 1), (k_1, 3), (k_2, 2), (k_2, 4)\}) = 6.$$

Thus, the total cost of the $\text{TA}(v_1)$ is four.

To summarize, the instance in Figure 2 is a worst-case instance for the $\text{CA}_{sum}$. □

**Theorem 7.** *The $\text{CA}_{sum}$ is a 7/5-approximation algorithm for $\text{CS}_{sum,s=t}$. Moreover, there exists an instance $I$ for which $\text{cost}(\text{CA}_{sum}(I)) = 7/5 \, \text{cost}(M^*(I))$.*

**Proof.** We assume w.l.o.g. that, for each $(k, \{i, j\}) \in M^*$, $\text{cost}(k, \{i, j\}) = w(d_k, s_i) + u_{ij}$. We have:

$$5\text{cost}(\text{CA}_{sum}) \leq 4\text{cost}(\text{MA}(1, u)) + \text{cost}(\text{TA}(v_1)) \tag{46}$$

$$\leq 4(v_3(M_3) + v_4(M_4)) + v_1(M_1) \tag{47}$$

$$\leq \sum_{(k, \{i,j\}) \in M^*} \left(4w(d_k, s_i) + 3w(d_k, s_j) + 4w(s_i, s_j)\right) \tag{48}$$

$$\leq \sum_{(k, \{i,j\}) \in M^*} \left(7w(d_k, s_i) + 7w(s_i, s_j)\right) \tag{49}$$

$$= \sum_{(k, \{i,j\}) \in M^*} 7 \, \text{cost}(k, \{i, j\}) \tag{50}$$

$$= 7 \, \text{cost}(M^*) \tag{51}$$

Inequality (47) follows from Lemma 5 and inequality (7). Inequality (48) follows from Lemmas 2 and 6. Inequality (49) follows from the triangle inequality. Inequality (50) follows from $\text{cost}(k, \{i, j\}) = w(d_k, s_i) + u_{ij}$.

We now provide an instance for which this ratio is achieved. Consider the instance $I$ depicted in Figure 3. This instance has four cars $\{k_1, k_2, k_3, k_4\}$ with car locations $\{d_1, d_2, d_3, d_4\}$ and eight requests $R = \{1, 2, \ldots, 8\}$. If two points are not connected by an edge, their travel time equals five. Observe that an optimal solution is

$$\{(k_1, \{1, 3\}), (k_2, \{2, 4\}), (k_3, \{5, 6\}), (k_4, \{7, 8\})\}$$

with $\text{cost}(M^*(I)) = 10$. Note that $M_R^* = \{\{1, 3\}, \{2, 4\}, \{5, 6\}, \{7, 8\}\}$. Let us now analyze the performance of $\text{MA}(1, u)$ and $\text{TA}(v_1)$ on instance $I$.

Based on the $u_{ij}$ values as defined in (20), the $\text{MA}(1, u)$ can find, in the first step, matching $M_3 = \{\{1, 2\}, \{3, 4\}, \{5, 6\}, \{7, 8\}\}$ because $v_3(M_3) = v_3(M_R^*) = 8$. Then, no matter how the second step assigns the request pairs to cars (since two cars stay at the same location), the total cost of $\text{MA}(1, u)$ will be 14.

$\text{TA}(v_1)$ may assign Requests 1 and 3 to Car 1 and Requests 2 and 4 to Car 2 and, similarly, Requests 5 and 7 to Car 3 and Requests 6 and 8 to Car 4 because:

$$v_1(\{(k_3, 5), (k_3, 7), (k_4, 6), (k_4, 8)\}) = v_1(\{(k_3, 5), (k_3, 6), (k_4, 7), (k_4, 8)\}) = 6.$$

Thus, the total cost of the $\text{TA}(v_1)$ is 14.

To summarize, the instance in Figure 3 is a worst-case instance for the $\text{CA}_{sum}$.  □

The $\text{CA}_{lat}$ runs the $\text{MA}(2, \mu)$ and $\text{TA}(v_2)$ and then outputs the better of the two solutions. We now state the main result for $\text{CS}_{lat}$. The following lemma is useful to analyze the performance of the CA for $\text{CS}_{lat}$.

**Lemma 8.** *For each* $(k, \{i, j\}) \in D \times R^2$,

$$\min\{2w(d_k, s_i, t_i) + w(t_i, d_k, s_j, t_j), 2w(d_k, s_j, t_j) + w(t_j, d_k, s_i, t_i)\}$$
$$+ 2w(d_k, s_i) + 2w(d_k, s_j) + \mu_{ij} + \mu_{ji}$$
$$\leq \min\{8w(d_k, s_i) + 5\mu_{ij}, 8w(d_k, s_j) + 5\mu_{ji}\}.$$

**Proof.** We first prove $2w(d_k, s_i, t_i) + w(t_i, d_k, s_j, t_j) + 2w(d_k, s_i) + 2w(d_k, s_j) + \mu_{ij} + \mu_{ji} \leq 8w(d_k, s_i) + 5\mu_{ij}$. We distinguish three cases based on $\mu_{ij}$: (1) $\mu_{ij} = w(s_i, t_i) + w(s_i, t_i, s_j, t_j)$; (2) $\mu_{ij} = w(s_i, s_j, t_i) + w(s_i, s_j, t_i, t_j)$; (3) $\mu_{ij} = w(s_i, s_j, t_j) + w(s_i, s_j, t_j, t_i)$.

Consider Case (1): $\mu_{ij} = w(s_i, t_i) + w(s_i, t_i, s_j, t_j)$. We have:

$$2w(d_k, s_i) + \mu_{ij} = 2w(d_k, s_i) + 2w(s_i, t_i) + w(t_i, s_j) + w(s_j, t_j).$$

According to the triangle inequality, we know that:

$$w(d_k, t_i) \leq w(d_k, s_i) + w(s_i, t_i),$$

$$w(d_k, s_j) \leq (d_k, s_i) + w(s_i, t_i) + w(t_i, s_j).$$

Based on the definition of $\mu$, we know:

$$\mu_{ij} = 2w(s_i, t_i) + w(t_i, s_j) + w(s_j, t_j),$$

$$\mu_{ji} \leq 2w(s_j, t_j) + w(t_j, s_i) + w(s_i, t_i) \leq 3w(s_j, t_j) + 2w(s_i, t_i) + w(t_i, s_j).$$

Using the above inequalities, we have:

$$2w(d_k, s_i, t_i) + w(t_i, d_k, s_j, t_j) + 2w(d_k, s_i) + 2w(d_k, s_j) + \mu_{ij} + \mu_{ji}$$
$$\leq 8w(d_k, s_i) + 10w(s_i, t_i) + 5w(t_i, s_j) + 5w(s_j, t_j)$$
$$= 8w(d_k, s_i) + 5\mu_{ij}.$$

The other two cases (2) and (3) are obtained similarly.

Analogously, we have $2w(d_k, s_j, t_j) + w(t_j, d_k, s_i, t_i) + 2w(d_k, s_i) + 2w(s_k, s_j) + \mu_{ij} + \mu_{ji} \leq 8w(d_k, s_j) + 5\mu_{ij}$.  □

**Theorem 8.** *The* $\text{CA}_{lat}$ *is a 5/3-approximation algorithm for* $\text{CS}_{lat}$.

**Proof.**

$$3\text{wait}(\text{CA}_{lat}) \leq 2\text{wait}(\text{MA}(2, \mu)) + \text{wait}(\text{TA}(v_2)) \tag{52}$$
$$\leq 2(v_3(M_3) + v_4(M_4)) + v_2(M_2) \tag{53}$$
$$\leq \sum_{(k, \{i,j\}) \in M^*} \min\{8w(d_k, s_i) + 5\mu_{ij}, 8w(d_k, s_j) + 5\mu_{ji}\} \tag{54}$$
$$\leq \sum_{(k, \{i,j\}) \in M^*} 5\min\{2w(d_k, s_i) + \mu_{ij}, 2w(d_k, s_j) + \mu_{ji}\} \tag{55}$$
$$= 5\,\text{wait}(M^*). \tag{56}$$

Inequality (53) follows from Lemma 5 and Inequality (8). Inequality (54) follows from Lemmas 4, 6, and 8.  □

**Theorem 9.** *The* $CA_{lat}$ *is a 3/2-approximation algorithm for* $CS_{lat,s=t}$. *Moreover, there exists an instance I for which* $\text{wait}(CA_{lat}(I)) = 3/2 \, \text{wait}(M^*(I))$.

**Proof.** We assume w.l.o.g. that, for each $(k,\{i,j\}) \in M^*$, $\text{wait}(k,\{i,j\}) = 2w(d_k,s_i) + w(s_i,s_j)$. We have:

$$2\text{wait}(CA(2,\mu)) \leq \text{wait}(MA(2,\mu)) + \text{wait}(TA(v_2)) \tag{57}$$

$$= \sum_{(k,\{i,j\})\in MA} \min\{2w(d_k,s_i) + \mu_{ij}, 2w(s_k,s_j) + \mu_{ji}\} \tag{58}$$

$$+ \sum_{(k,\{i,j\})\in TA(v_2)} \min\{2w(d_k,s_i) + \mu_{ij}, 2w(s_k,s_j) + \mu_{ji}\} \tag{59}$$

$$= v_3(M_3) + v_4(M_4) + v_2(M_2) \tag{60}$$

$$\leq \sum_{(k,\{i,j\})\in M^*} \left(w(s_i,s_j) + w(d_k,s_i) + w(d_k,s_j) + 3w(d_k,s_i) + w(d_k,s_j)\right) \tag{61}$$

$$\leq \sum_{(k,\{i,j\})\in M^*} \left(3w(s_i,s_j) + 6w(d_k,s_i)\right) \tag{62}$$

$$= \sum_{(k,\{i,j\})\in M^*} 3 \, \text{wait}(k,\{i,j\}) \tag{63}$$

$$= 3 \, \text{wait}(M^*). \tag{64}$$

Equation (59) follows from (23). Equation (60) follows from Lemma 5. Inequality (62) follows from the triangle inequality. Equation (63) follows from $\text{wait}(k,\{i,j\}) = 2w(d_k,s_i) + \mu_{ij}$.

We now provide an instance for which this ratio is achieved. This instance has four cars $\{k_1,k_2,k_3,k_4\}$ with car locations $\{d_1,d_2,d_3,d_4\}$ and eight requests $R = \{1,2,\ldots,8\}$. Consider the instance $I$ depicted in Figure 5. If two points are not connected by an edge, their travel time equals five. Observe that an optimal solution is:

$$\{(k_1,\{1,3\}), (k_2,\{2,4\}), (k_3,\{5,6\}), (k_4,\{7,8\})\}$$

with $\text{wait}(M^*(I)) = 8$. Note that $M_R^* = \{\{1,3\},\{2,4\},\{5,6\},\{7,8\}\}$. Let us now analyze the performance of the $MA(2,\mu)$ and $TA(v_2)$ on instance $I$.

**Figure 5.** A worst-case instance for the $CA_{lat}$ of $CS_{lat,s=t}$.

Based on the $\mu_{ij}$ values as defined in (22), the $MA(2,\mu)$ can find, in the first step, matching $M_3 = \{\{1,2\},\{3,4\},\{5,6\},\{7,8\}\}$ because $v_3(M_3) = v_3(M_R^*) = 4$. Then, no matter how the second step matches the pairs to cars, the total waiting time of the $MA(2,\mu)$ will be 12.

The $TA(v_2)$ may assign Requests 1 and 3 to Car 1 and assign Requests 4 and 2 to Car 2 and, similarly, assign Requests 5 and 7 to Car 3 and assign Requests 6 and 8 to Car 4 because:

$$v_2(\{(k_3,5),(k_3,7),(k_4,6),(k_4,8)\}) = v_2(\{(k_3,5),(k_3,6),(k_4,7),(k_4,8)\}) = 8.$$

Thus, the total waiting time of the $TA(v_2)$ is 12.

To summarize, the instance in Figure 5 is a worst-case instance for the combined algorithm $CA_{lat}$. $\square$

## 5. Conclusions

We analyzed a polynomial-time algorithm, called the transportation algorithm (TA), for four different versions of a ride-sharing problem where each car serves at most $c$ requests. We proved that the TA is a $(2c-1)$-approximation (resp. $c$-approximation) algorithm for $CS_{sum}$ and $CS_{sum,s=t}$ (resp. $CS_{lat}$ and $CS_{lat,s=t}$). Furthermore, for the special case where capacity $c = 2$ and $m = c \cdot n$, we proposed another algorithm, called match-and-assign (MA), which firstly matches the requests into pairs and then assigns the request pairs to the cars. We proved that (for most problem variants) the worst-case ratio of the algorithm defined by the better of the two corresponding solutions is strictly better than the worst-case ratios of the individual algorithms.

For future directions, it would be interesting to extend the MA for the ride-sharing problem for any constant capacity $c$. It would also be interesting to obtain meaningful lower bounds on the approximability of the ride-sharing problem. Other possible directions include studying the problem under different objectives such as minimizing the makespan or considering the release times and/or deadlines of the requests.

**Author Contributions:** Investigation, conceptualization, methodology, formal analysis, K.L. and F.C.R.S.; writing—original draft preparation, K.L.; writing—review and editing, F.C.R.S. All authors have read and agreed to the published version of the manuscript.

**Funding:** This project has received funding from the European Union's Horizon 2020 research and innovation programme under the Marie Skłodowska-Curie grant agreement number 754462 and funding from the NWO Gravitation Project NETWORKS, Grant Number 024.002.003.

**Institutional Review Board Statement:** Not applicable.

**Informed Consent Statement:** Not applicable.

**Data Availability Statement:** Not applicable.

**Conflicts of Interest:** The authors declare no conflict of interest.

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
