# Peer review of "Minimizing Travel Time and Latency in Multi-Capacity Ride-Sharing Problems"

_algorithms, doi:10.3390/a15020030_

Round 1

Reviewer 1 Report

The article presents methods for solving operational problems in ride-sharing.
The solutions are based on solving the assignment (minimum-cost matching) problem in graphs with adequately chosen edge weights. 
It is proven that the solutions thus obtained have provable approximation ratios within a constant factor of optimal.

In general, the paper is well written, the arguments are clearly presented, and the problem itself is well-motivated and interesting.
I have a few suggestions that I would appreciate the Authors take into consideration when revising the paper.

  • Perhaps the biggest issue for me:
    In Algorithm 1 on p. 6, the authors give an input parameter v in {v_1, v_2}, but at this point, v_1 and v_2 are not defined.
    Then, v_1 and v_2 themselves are calculated in the algorithm.
    Perhaps the Authors aimed to achieve the same format as in Algorithm 2 on p. 11, where two cost functions u and mu are appropriately defined beforehand, and the operations in the algorithm are independent of the choice of a cost function.
    - My honest and strong suggestion:
    Eliminate the pseudo-code of Algorithms 1 and 2.
    Instead, make an argument that for an appropriately constructed graph and edge function, the cost of a minimum-cost perfect matching in the graph has [the desired property] with respect to [the original cost function for the problem].
    Then, leave it to the reader to imagine the two for-loops necessary to construct such cost functions, and the one line in pseudocode that says "Find a minimum weight perfect matching."
    In fact, if Algorithm 1 is properly written as Algorithm 2 is with the cost functions v_1 and v_2 appropriately defined beforehand, then lines 3-12 in the pseudo-code become obsolete, and we end up with an algorithm with three lines:
    1. construct a graph
    2. find a min-weight assignment
    3. output.
    I hope the Authors would agree that the reader does not need an outlined pseudocode to get this.
    - (Minor issue with this respect, please commit to either "assignment" or "matching." I know that those terms are equivalent, but I find it a bad habit to spice things up by using synonyms in technical writing that should be otherwise precise.)
  • Other comments:
    • p.1, line 22: is it necessary to assume constant velocity at this point?
      One can always assign edge weights that represent travel times instead of distances. Seems like unnecessary acrobatics in assumptions and problem modeling.
    • p.2, line 70: "A regular car has at most four available seats..."
      I beg to differ, it is quite common these days for ride-shares (in the US, Japan...) to be 6-8 seaters, which as private cars are regular, ordinary type of cars.
      I realize that his fact has almost no impact on the theoretical arguments of the paper, it might raise the constant c from 2 to 3, which is still irrelevant.
      However, such a bluntly put assertion here just begs to be contradicted.
    • p. 2, line 74: TransVision provide -> TransVision provides...
    • p. 2, line 76: combines these requests -> combine these requests...
    • p. 2, line 77: to some regular transport companies.
      "Regular" in the sense that they use "regular" cars with at most four available seats? What makes TransVision irregular?
    • p. 2, line 79: 5.000 : I know that continental Europe uses a decimal comma and a dot separator for thousands, but this might cause some confusion for the rest of the world.
    • p. 3, line 96: [6, 7, 10], [5] -> [5-7, 10]
    • p. 3, line 132-133: "that satisfy the triangle inequality". I found it strange that a function defined over triplets should satisfy the triangle inequality, but in the cited reference it is in fact the weights d that are subject to this assumption.
      Not much relevant to the rest of the paper, but this passage should be improved.
    • p. 3, line 134-135: this expression: what is "this" here?
    • p. 4, line 147 (and elsewhere): the best of the solutions.... -> the better of (in this case two), save the best for when there are three or more.
    • p. 4, line 158: "Given a metric space on vertices V." This is usually a sub-clause in a complex statement, and is meaningless by itself, as it has no main verb.
    • p. 4, line 171: "instance" -> case. I believe that instance should refer to a particular problem instance with fixed parameters, whereas a case covers a particular set of instances.
    • p. 4-5. Definition of the function SHP.
      This involves the cost of a shortest Hamiltonian path. Theoretically, not a problem, but one must wonder whether such a cost function can be used in practice, since the  Shortest Hamiltonian Path is an NP-hard problem.
      I realize that since the parameter c is assumed to be a small constant, this is not an issue, but a gentle reminder for the sleepy reader would be nice here.
    • p. 5, definition of SHWP. There is not a definition to speak of, merely some intuitive description. I would appreciate a clear, stand-alone definition.
    • p. 5, line 194. As mentioned above, there is no point in defining the cost functions v_1 and v_2 inside an algorithm.
      Even better, you can eliminate the algorithms, as is my strong suggestion.
    • p. 6, Algorithm 1:
      (Again, better eliminate this algorithm... but, some points to consider)
      is it necessary to give the set P of virtual car sets in the input of the algorithm? As it is unambiguously defined from the set D of cars, is it really a degree of freedom necessary to be given as input?
    • p. 6, lines 206, 207 (and elsewhere): lemma's -> lemmas or lemmata (the accepted plural forms of lemma).
    • p. 7, lines 234-240: I believe the math expressions can be more comfortably read if written in display mode as opposed to inline mode.
    • p. 10, lines 317-320: We claim that...
      A short hint as to why such a claim is true would be appreciated here.
      Since all the dummy weights are 0, such an explanation should be quite short, yet helpful here.
    • p. 10, lines 327-328: For convenience, we replace...
      I believe that the "replacement" follows by the definition of SHP, it is a brute-force write-up of the minimum cost of a Hamiltonian path.
      Instead of "replace", the Authors might say that they explicitly write the quantity...
    • p. 14, last line of the proof of Theorem 4: a small comment explaining the tightness of the instance in Fig. 4 would be appreciated.
    • p. 17, lines 435-440: Please use display mode for readability.
    • p. 19, line 477-479: I am completely lost with the intention of this sentence. I would appreciate the Authors to elaborate in a few more lines their intended plans for future research.

Last, a small, yet I would appreciate if it is heeded, comment on the use of English:
- p. 3, line 139: ... is reminiscent of a type of heuristics...
- p. 4, lines 145-146: ... the presence of algorithm MA allows...
I strongly suggest the co-author that crafted the fine examples of prose above to check the second bullet point on page 18 of Jordan Peterson's "Essay Writing Guide."
Coupled with the slight typos and comments on the verb form for the third person in present simple singular, I believe the Author would get my feeling when I read this sentence. 

Author Response

Thank you very much for your comments. All of the suggestions are very helpful. We provide a response to each of the comments and attach the revised version. Please see the attachment. 

Reviewer 2 Report

The authors present results of their previous conference
paper on the subject [19] together with a generalization.

The article is well written, but some details could be
added for better readability.
For example, there is a minus sign in the Lemma 1, but nowhere
this minus sign appear in Algorithm 1; thus an explanation 
could be added to explain where it comes from.

Author Response

Thank you very much for your comments. We provide a response to the comment and attach the revised version. Please see the attachment. 

Reviewer 3 Report

Review of the article: “Minimizing travel time and latency in multi-capacity ride-sharing problems”

The article analyzes two algorithms: the polynomial algorithm, called the transport algorithm (TA), for four different versions of the common travel problem, and the match-and-assign (MA) algorithm.

General conclusion: The article deals with the current research topic. It is well written but incomplete. Hence, it requires some corrections before possible publication.

As it was written in the general conclusion, the presented article in the presented form (description of algorithms) is well and clearly written. The presented theory is of a high scientific level. And I have no critical comments on this part. However, the reviewer lacks some parts in the article that are normally found in this type of article:

  • The article does not indicate the research gap that is solved in it.
  • There is no clear comparison between the presented algorithms and the existing ones, e.g. indicating the advantages of own algorithms.
  • There is no chapter in which the effectiveness of proprietary algorithms could be compared to the existing ones using test examples.
  • The literature review contains few items and most of them are quite old.

Hence, in the presented form, the reader does not get any arguments to consider the use of the advanced algorithms presented here in his analyses.

Author Response

(The authors gave the same response as above.)

Round 2

Reviewer 1 Report

I am grateful to the Authors for taking into account my comments and preparing a thoughtful reply letter.

I believe that the manuscript can be accepted in its current form.

Reviewer 3 Report

The present version of the manuscript has been significantly changed and improved. The manuscript could be considered suitable for the publication.